# Characteristics and Patient Reported Outcome Measures in Lipedema Patients—Establishing a Baseline for Treatment Evaluation in a High-Volume Center

**DOI:** 10.3390/jcm11102836

**Published:** 2022-05-17

**Authors:** Matthias Hamatschek, Henning Knors, Marie-Luise Klietz, Philipp Wiebringhaus, Matthias Aitzetmueller, Tobias Hirsch, Maximilian Kueckelhaus

**Affiliations:** 1Department of Plastic, Reconstructive and Aesthetic Surgery, Hand Surgery, Fachklinik Hornheide, 48157 Muenster, Germany; hamatschek@googlemail.com (M.H.); hknors@aol.com (H.K.); marie.luise.klietz@gmx.com (M.-L.K.); philippwie@gmx.de (P.W.); aitzetmueller.m@hotmail.com (M.A.); tobias.hirsch@uni-muenster.de (T.H.); 2Plastic and Reconstructive Surgery, Institute of Musculoskeletal Medicine, Westfalian Wilhelms-University, 48149 Muenster, Germany; 3Division of Plastic and Reconstructive Surgery, Department of Trauma, Hand and Reconstructive Surgery, University Hospital Muenster, 48149 Muenster, Germany

**Keywords:** lipedema, liposuction, QoL, depression, obesity, restrictions in daily life

## Abstract

Lipedema patients suffer not only from visual stigma but also reduction in their quality of life through pain and performance loss in daily life. In clinical practice, it is still difficult to reliably diagnose the disease. This study aims to provide further insights into the characteristics of lipedema patients of all stages and provide a baseline prior to surgery for a surgical treatment evaluation by means of patient-reported outcome measures. Methods: Patients completed a lipedema-specific questionnaire containing 50 items, the World Health Organization Quality of Life BREF (WHOQOL-BREF) and the Patient Health Questionnaire 9 (PHQ-9). The data were analyzed using SPSS statistics 27. Patients who had already received liposuction were excluded. Results: Five hundred and eleven patients were included, of whom 337 completed the PHQ9 and 333 completed the WHOQOL-BREF questionnaires. The general characteristics of lipedema patients, especially the daily symptoms, are described. Previous observations, such as the frequent occurrence of hypothyroidism and the low rate of type 2 diabetes, were confirmed. Over 49% suffer from severe impairments in their jobs, whereby the disease shows a familial accumulation. The results of the WHOQOL-BREF and the PHQ-9 suggest a high level of mental stress. Discussion: As surgical intervention in lipedema patients is gaining traction, its effects should be well-documented. Therefore, a comprehensive baseline needs to be established prior to surgical treatment. The psychological components are just as important as the inclusion of daily impairments.

## 1. Introduction

Lipedema is a potentially widespread disease appearing almost exclusively in women. It is often misdiagnosed as obesity [1,2] while leading to high depression levels and significant impairment on their quality of life (QoL) [3].

The reported prevalence varies between 5 and 15% in women [4,5,6]. Patients frequently have to visit more than three doctors, and often, more than 15 years pass prior to a correct diagnosis [7,8]. In a study from the United Kingdom, 54% of district general hospital consultants did not recognize or had even seen this disease in their everyday professional life [9].

A symmetric accumulation of painful adipose tissue, especially in the limbs, is characteristic and leads to a disproportionate tissue distribution [1,2,10]. While the legs are almost always affected, the arms are more often affected in higher stages [4,11,12]. Pain in the adipose tissue is one of the most common symptoms, which causes a lot of limitation in everyday and professional life [12]. Patients report additional symptoms in the affected areas, such as bruising, sensitivity to touch, feeling of tension, heavy and tired limbs and aesthetic impairment (Table 1) [13,14,15,16]. Affected patients usually recognize the first symptoms like enlargement of the legs during adolescence [3,7,8,17].

In 2004, Meier-Vollrath and Schmeller developed a systematic classification by subdividing the disease into three stages, depending on the visual and haptic findings of adipose tissue [11,12,18]. In addition, lipedema is divided into five subtypes, depending on the primary affected region [4,19] (Table 1).

**Table 1 jcm-11-02836-t001:** Types of lipedema and diagnostic criteria of lipedema [1,2,19].

Diagnostic Criteria	Type of Lipedema	Location of Onset
Almost exclusive occurrence in women	1	Hips
Bilateral and symmetrical manifestation with minimal involvement of the feet	2	Hips and thighs
Persistent enlargement after elevation of the extremities or weight loss Arms are affected	3	Hips, thighs and shanks
Minimal pitting edema	4	Arms
Negative Kaposi–Stemmer sign	5	Shanks
Pain, tenderness on pressure		

Most lipedema patients receive a noninvasive, conservative therapy [20]. Complex decongestive therapy (CDT) consists of manual lymphatic drainage (MLD), intermittent pneumatic compression (IPC) and a compression garment.

Not only during hospitalization but also in outpatient treatment, patients benefit from CDT through leg volume and pain reduction [4,21,22]. However, these outcomes are frequently diminished during the outpatient periods. Some patients overcompensate for these effects so that their leg volume increases overall [21]. The compression garment needs to be worn as much as possible in order to reduce edema and pain in everyday life in over 50% of patients [20].

The current literature suggests that liposuction may be a promising approach as an invasive treatment modality in terms of pain reduction and increase in the quality of life [13,14,15,16,23].

Overall, the outcome data of liposuction in lipedema are sparse, and studies in patients with a confirmed lipedema diagnosis are rare.

With this study, we aim to gain further insight into the characteristics of lipedema patients of all stages. In addition, to evaluate surgical treatment, we provide a baseline before surgery for patient-reported outcomes (PROMs).

## 2. Methods

This study was approved by the local ethics committee. Ethic Committee Name: Ethics Committee of the Medical Association Westphalia-Lippe and Westphalian Wilhelms University. Approval Code: 2021-684-f-S. Approval Date: 9 March 2022.

### 2.1. Patient Selection

All patients visiting our institution with the suspected diagnosis of lipedema were included in the study. Each patient was asked to fill in an individually designed questionnaire prior to their consultation (Appendix A). Patients who already received a liposuction or any bariatric surgery prior to consultation, were a minor or who turned out to not be suffering from lipedema were excluded.

### 2.2. Structure of Questionnaire

General information such as height, weight, age of symptom onset, age at diagnosis, location of pain and family history of lipedema were evaluated. Patients were asked to give information about their impairment in professional life and which therapies they had undergone so far. Further, comorbidities and smoking behavior were assessed.

Twenty questions referred to the patients’ pain symptoms using a numeric rating scale from one to ten. The subjects were: exact location of pain, bruising, swelling, heat and coldness of extremities, muscle cramps, heaviness or fatigue of the legs, skin problems, pruritus, limitations while walking and in quality of life and satisfaction with optical appearance of the legs.

The QoL and level of depression were monitored via the World Health Organization Quality of Life-BREF (WHOQOL-BREF) and the Patient-Health-Questionnaire-9 (PHQ-9).

The WHOQOL-BREF is a patient-reported outcome measurement tool to monitor their overall health. It contains twenty-four items divided into four domains to measure every facet of their quality of life using a Likert scale from one to five. These health domains are physical health, psychological health, social relationships and environment. Further, two single “benchmark” items are given to monitor the general aspects of general health and overall QoL. This questionnaire shows high validity in studies to map the quality of life [24,25,26,27].

The PHQ-9 was applied for the depression severity evaluation. It consists of nine different items assessing depression. Participants were asked to provide the frequency of symptom appearance during the last 28 days via a four-point-Likert-scale (from zero (=not at all) to four (=nearly every day)). The higher the score, the higher the depression severity. A score of five to nine equals “minimal symptoms”, while 20 and more points demonstrate “serve depression” [28,29].

### 2.3. Data Collection

Data were collected in an individually designed database in SPSS statistics 27. A retrospective analysis was performed. Spearman’s correlation coefficient was determined to show positive and negative correlations. For a confirmatory analysis, an analysis of variance (ANOVA) was performed to show differences between groups due to continuous outcome variables. Due to multiple testing, a Bonferroni correction was used to adjust the global significance of 5%. Each test was performed to a local significance level of 1%.

## 3. Results

### 3.1. General Information

Five hundred and eleven patients met the eligibility criteria and were included in this study. Patients had a mean age of 40.16 (±12.45) years and a BMI of 33.31 kg/m^2^ (±7.8) across all stages. The lipedema stage and BMI demonstrated a positive correlation (Spearman r = 0.566, ANOVA *p* < 0.001) (Table 2). On average, 16.11 (±11.39) years elapsed between the symptom onset and diagnosis. Table 2 presents detailed information on the patient characteristics.

Over 80% of patients noticed the first increased tissue proliferation during hormonal changes, such as puberty (67.3%), pregnancy (9.5%) and during menopause (4.0%).

At disease onset, 50.1% of patients had both thighs and lower legs affected, i.e., the entire leg. On the day of presentation in our institution, the entire leg was affected by increased fat accumulation in 82.4% of patients (Figure 1).

If we look at the dietary behaviors of the patients, we see that only 7% had never tried to lose weight by changing their diet. In contrast, 25% said that their efforts of dietary changes did not show any affect, whereby 44.5% had little success. About 41% declared that they were able to lose up to five kilograms of body weight.

### 3.2. Comorbidities

Table 2 gives detailed information about the occurring concomitant diseases. There was shown that 31% of patients suffered from hypothyroidism. Adiposity-associated diseases like hypertonia and type 2 diabetes occurred with a frequency of 18.5 and 3%. In Figure 1, the concomitant diseases are also differentiated according to stage. The number of patients with hypertension increased to 30.7% at stage 3.

In addition, joint problems were more prevalent in stage 3 patients (39.8%, *n* = 64 of 161) than in stage 1 patients (11.9%).

Thrombosis also occurred less often in patients with stage 1 (4.9%) than with stage two (10.3%) or three (10.1%).

### 3.3. Familial Clustering

Patients were asked if any family members suffered from lipedema or remarkably “thick legs”.

In 33.6% of cases, the patient’s mother was also suspected to be affected by lipedema. With 28.4%, grandmothers were suspected to be affected in the previous generation (maternal 15.4%, paternal 10.3%). Further detail is shown in Table 3 and visualized in Figure 2.

### 3.4. Symptoms

Eighteen different items were asked by means of a numerical rating scale from one (none) to ten (very serve/strong) to establish a daily symptom overview (Table 2). The highest mean scores were not for leg pain (6.68 (±2.32)) but for tiredness (7.79 (±2.17)) and heavy legs (8.21 (±1.95)). Likewise, hematomas (7.63 (±2.31)) and the feeling of tension (7.49 (±2.19)) were prominent in the affected regions (Table 2).

### 3.5. Effects on Daily Life

During work 96% of patients suffered from leg impairment, while 64% also had problems in their arms. Figure 3 illustrates the restrictions in detail. These complaints led to occupational restrictions of varying severity. Over all the stages, 49% suffered from severe impairment in their jobs or were even completely unable to work. Figure 3 shows a differentiation by the stages.

On a numeric rating scale from one to ten regarding the quality of life reduction (one = no reduction, ten = very severe reduction) across all stages, the most picked single score (25.33%) was the maximum score of ten (Figure 4).

There was a statistically significant, positive correlation between the stage of lipedema and the level of the numerical rating scale score (Spearman r = 0.55, ANOVA *p* < 0.001) (Figure 5a).

### 3.6. Mental State

The PHQ-9 was fully completed by 337 patients. In total, the mean value was 10.84 (±6.39). In sum, the test suggested that 54.01% of lipedema patients were at risk of suffering from a moderate to severe depressed mood. The higher the lipedema stage was, the higher the PHQ-9 score was (Spearman r = 0.2, ANOVA *p* = 0.002) (Figure 5b).

There was a highly significant positive correlation between the degree of occupational disability and the PHQ-9 score (Spearman r = 0.413, ANOVA *p* = <0.001) (Figure 5d).

Three hundred and thirty-three patients completed the WHOQOL-BREF questionnaire. The mean value of all the domains was 60.5 (±16.02). The lowest scores were obtained in the physical (mean 54.54 ± 20.1) and mental (mean 51.91 ± 18.67) health domains, while the highest mean score was obtained in the environment domain (mean 71.85 ± 16.00).

The WHOQOL-BRREF score had a highly significant negative correlation with the degree of occupational limitation (Spearman r = −0.406, ANOVA test *p* < 0.001) (Figure 5c).

Patients who had a higher BMI also showed significantly lower scores in the WHOQOL-BREF questionnaire (r = −0.353, ANOVA *p* = 0.016).

There was a significant negative correlation between the PHQ-9 and the WHOQOL-BREF scores (Spearman r = −0.775, ANOVA *p* < 0.0001).

**Figure 5 jcm-11-02836-f005:**
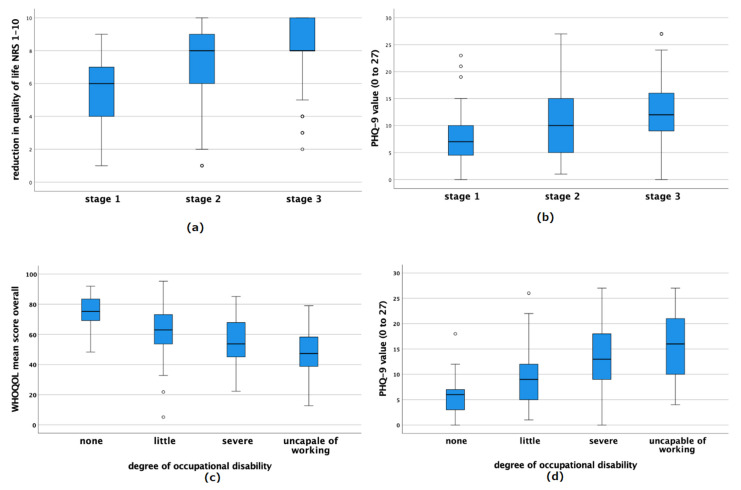
The quality of life was assessed and correlated: (**a**)—the reduction in quality of life differentiated by stages is shown (ANOVA test *p* < 0.0001; Spearman r = 0.55 *p* < 0.000); (**b**)—the PHQ-9-score differentiated by stages is shown (Spearman r = 0.201 *p* = < 0.001; ANOVA test: *p* = 0.002); (**c**)—the relation between the quality of life and degree of occupational disability (Spearman r = −0.406 *p* < 0.001; ANOVA test *p* < 0.001); (**d**)—relation between the PHQ-9 value and the degree of occupational disability (Spearman r = 0.413, *p* < 0.0001; ANOVA, *p* < 0.0001).

## 4. Discussion

Lipedema is a multifaceted disease that affects patients’ lives in multiple ways. This study characterizes a large monocentric lipedema patient cohort prior to surgery, combining clinical data with validated HRQOL and depression questionnaires. The presence of the disease was confirmed by clinicians, and the stage was professionally determined according to the valid criteria.

Setting this baseline is crucial in order to accurately record and evaluate the effectiveness of liposuction for lipedema treatment, especially from the patient-reported outcome perspective.

The often-publicized observations that the disease initially becomes symptomatic during hormonal upheavals and multiple years pass from disease onset until correct diagnosis are supported by our findings [7,8,17,30].

One of the most serious symptoms of lipedema patients in our study, besides pain, was the tendency for frequent hematomas in the affected regions. As a possible explanation for this, a higher capillary fragility (CF) was observed with angiosterrometry using the Parrot’s Angiosterrometer in 2008 [22].

The most commonly reported comorbidities, according to the current literature, is hypertension (13–27%) and hypothyroidism (20–35% or higher) [8,11,17,31]. Additionally, these observations were confirmed by our data. The clustered occurrence of hypothyroidism is particularly interesting, as this may promote obesity [32,33]. As demonstrated, the BMI showed a highly significant positive correlation with lipedema progression. The increased accumulation of fatty tissue on the legs makes this a logical consequence. It should be critically questioned whether the sole determination of the BMI can provide a valid statement about the obesity status. Compared with obese populations, lipedema patients showed a lower prevalence of type 2 diabetes and hypertension [34,35].

The disease may be masked by concomitant obesity, making it difficult for the clinician to differentiate. Further studies may also elucidate the causal relationship between lipedema and patients’ hormonal status. Here, another discriminant could be determined to increase the discriminatory power between lipedema and obesity.

As our data showed, more than a quarter of patients already unsuccessfully tried to lose weight. By losing about five kilograms of body weight, 40% had only limited success.

Recent publications have discussed the positive effects of special diets, such as Ketogenic or Mediterranean diets, on the body composition of lipedema patients [36,37]. As part of a multimodal treatment concept, more intensive nutritional medical consultations and individual discussions of the appropriate dietary form seem necessary. Patients should be offered professional sports counseling to reduce adipose tissue that is still reactive to dietary changes and exercise. Likewise, it could be necessary to analyze the body composition in a more detailed way. By using a bioimpendance analysis (BIA) quantification of fat accumulation, distribution, as well as monitoring it over the course of the therapy, is possible. In addition, it may be possible to identify differences between lipedema and obesity patients, which could be a diagnostic criterion and a distinctive feature in the future. This method is already widely used and could be applied in practice with little effort [38,39,40,41].

The current literature has already reported a familial clustering of the disease and suggested a genetic disposition [8,42,43]. Our data confirmed this clustering, especially among the patients’ mothers and grandmothers. A bias in self-reporting cannot be excluded, because there may be an increased awareness for this disease in families where lipedema occurs. Nevertheless, it may be possible that the frequent occurrence in families offers a further diagnostic criterium to distinguish lipedema from obesity.

Lipedema patients show a high potential to develop depression or already suffer from depression. This is shown by increased scores in the PHQ-9 index and is also reflected in a significant quality of life reduction in lipedema patients. An increased 12-month prevalence for depression at 25% compared to about 8% in the general population has already been demonstrated [11,15,30]. We confirmed these findings with data from our large heterogeneous lipedema population.

The common practice of assessing the QoL or personal well-being with single-item questionnaires should be abandoned in future publications and clinical practice. By assessing the mental state of patients with valid and multimodal questionnaires, a comprehensive documentation can be obtained, with the QoL and the valid evaluated level of depression as valuable parameters.

Concerning surgical treatment options, not only with regards to the near-term reduction of adipose tissue and body weight but also with the long-term results, liposuction offers promising results so far. It may reduce the level of pain and general limitations in everyday or professional life and increase the quality of life and even the subjective quality of patients’ sex lives [8,11,13,14,15,16,23,31,44,45,46]. Furthermore, it reduces the need for CDT in about 46% of patients, and a significant reduction (*p* = 0.0002) of the number of migraine attacks per month was observed after liposuction [11,14].

However, to date, there has been only one data series that monitored the same patient population several times over a long period of time [14]. In addition, some of the current data were collected retrospectively, meaning that patients had to recall their condition before their surgery [15]. A recall bias was excluded in our data, since patients self-reported their complaints prior to their first liposuction. In addition, the current literature lacks validated questionnaires on the quality of life and the psychological state of the patients. With a standardized procedure for PROMs, it will further be possible to determine whether WAL or TLA is the best method for treating lipedema. Taking this into consideration, we will be able to adequately monitor our treatment progress and be able to present the effectiveness of liposuction with high-quality data. As the socioeconomic impact of lipedema has been underrated for a long time, providing extensive high-quality data will not only serve to provide best possible treatment for patients but also ensure the recognition of the surgical treatment concept and its compensation.

In conclusion, lipedema remains a poorly understood disease. This study provided insight on family clustering, clinical characteristics and health-related quality of life in patients prior to surgical treatment and thus provided a baseline for the outcome assessment. This evaluation will be crucial for the improvement of multimodal treatment concepts and thus sustainably improve the quality of treatment for patients.

## Figures and Tables

**Figure 1 jcm-11-02836-f001:**
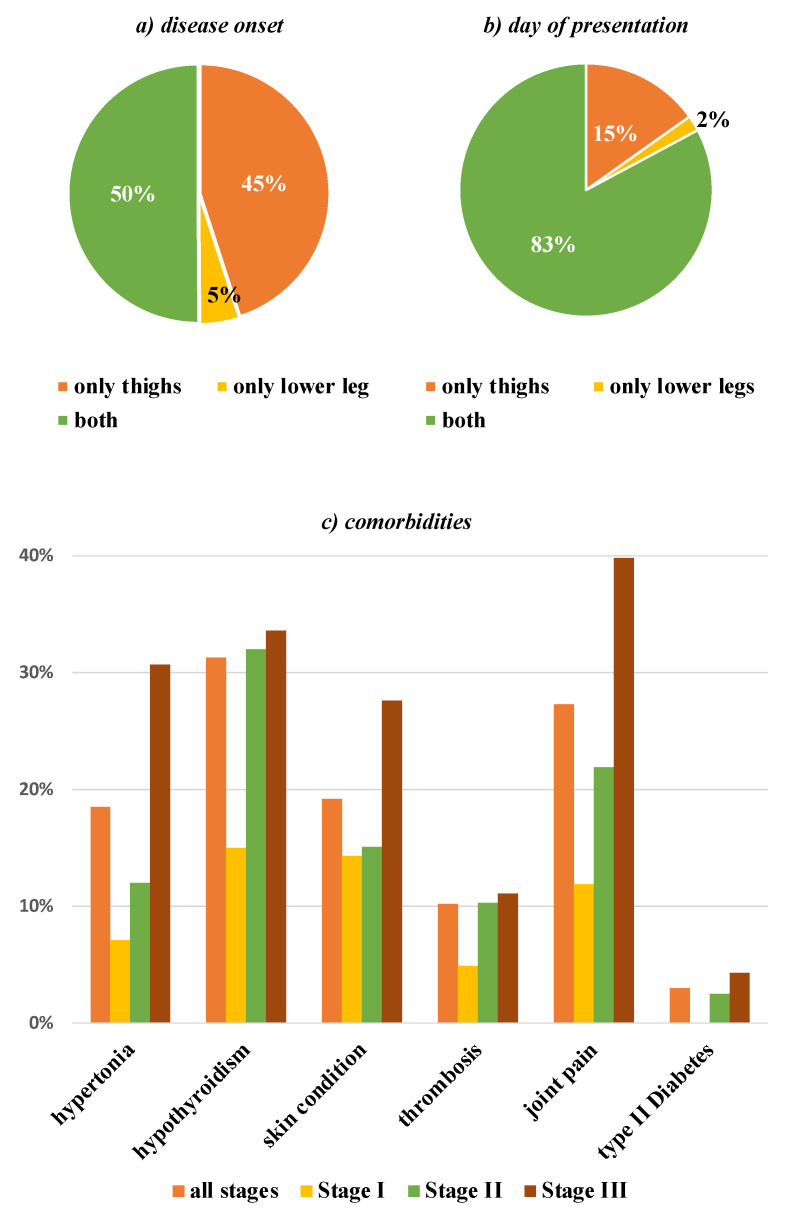
Patient characteristics: (**a**) shows the affected regions at the time of disease onset. In comparison, (**b**) shows the affected regions at the time of presentation to our clinic. (**c**) illustrates the comorbidities of lipedema patients, differentiated according to their present stage.

**Figure 2 jcm-11-02836-f002:**
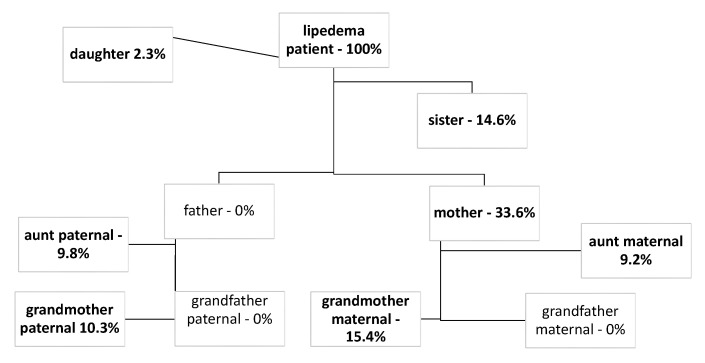
The pedigree of familial occurrence of enlarged legs shows the frequency within the immediate family.

**Figure 3 jcm-11-02836-f003:**
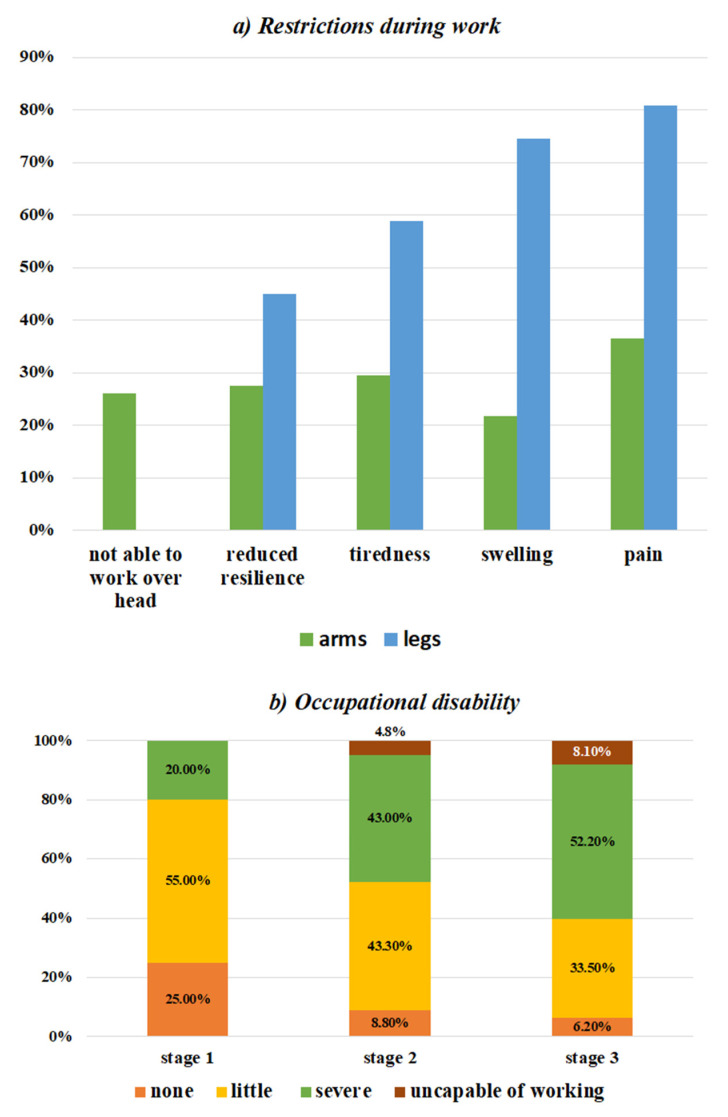
Restrictions and disability at work: (**a**) gives a detailed look at the restrictions during work, distinguishing whether arms or legs are affected, and (**b**) shows the degree of occupational disability differentiated by stage.

**Figure 4 jcm-11-02836-f004:**
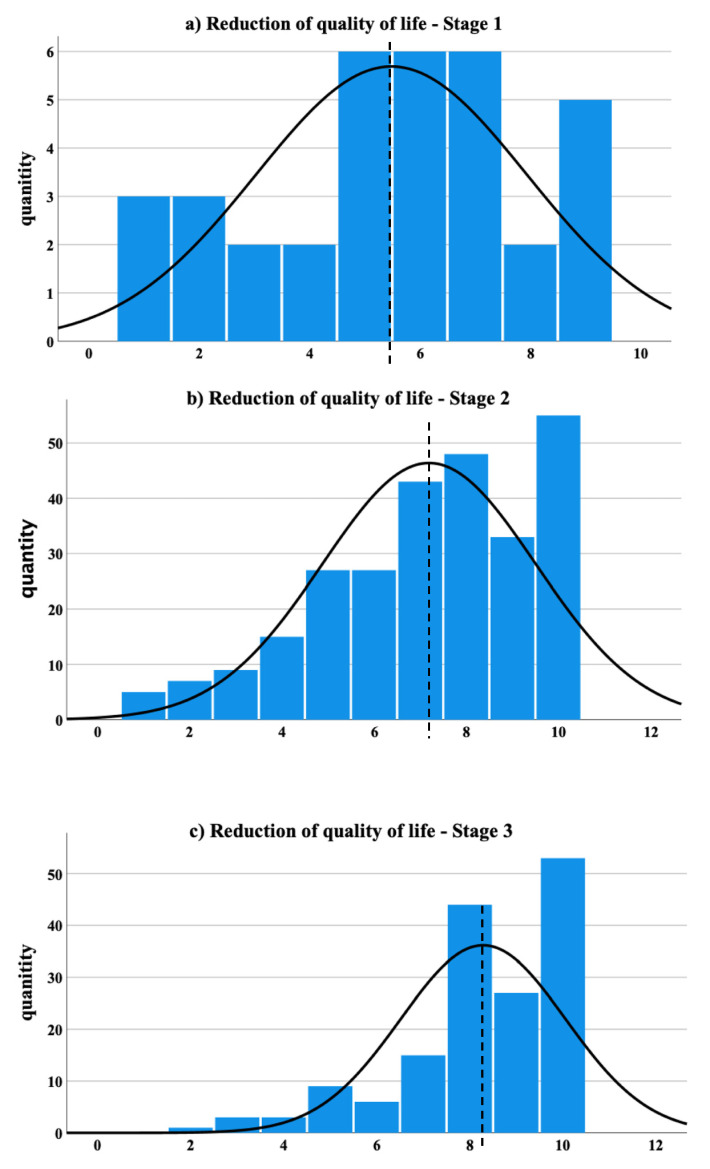
The reduction of the quality of life, by means of a numerical rating scale from one to ten, is differentiated by stage. The results for Stage 1 are shown in (**a**) (mean 5.49 ± 2.55), whereby Stage 2 is shown in (**b**) (mean 7.19 ± 2.13), and Stage 3 is shown in (**c**) (mean 8.29 ± 1.78). The dashed line illustrates the mean value. The solid curved line represents the normal distribution.

**Table 2 jcm-11-02836-t002:** Patient characteristics (population *n* = 511).

Age at Presentation	Mean 40.16 (±12.45)
Weight at presentation	mean 96.16 kg (±23.11)
BMI at presentation	mean 33.13 (±7.8)
BMI < 18.5	0.0%
BMI 18.5 to 24.99	11.7%
BMI 25 to 29.99	23.2%
BMI 30 to 34.99 (obesity I°)	24.2%
BMI 35 to 39.99 (obesity II°)	19.6%
BMI > 40 (obesity III°)	20.2%
Age at onset of symptoms	mean 19.66 (±10.00)
Age at diagnosis	mean 36.69 (±11.79)
years between onset and diagnosis	mean 16.11 (±11.39)
BMI stage I	mean 24.71 ± 3.61
BMI stage II	mean 32.11 ± 6.00
BMI stage III	mean 39.33 ± 7.53
Stage I legs (at presentation)	8.6%
Stage II legs	57.8%
Stage III legs	33.6%
Stage I arms	11.8%
Stage II arms	36.0%
Stage III arms	9.9%
Stage 0 arms	42.3%
Smoking behavior	87.2%—nonsmoker 6.4%—<5 per day 3.4%—5–10 per day 1.8%—11–15 per day 1.2%—16–20 per day
Comorbidities	75.9%
hypothyroidism	31.3%
joint pain	27.3%
Skin problems	19.2%
Hypertonia	18.5%
	10.2% patient
thrombosis	9.3% only family member
Type 2 diabetes	3.0%
Symptoms	mean (±SD) (*n* = 511)
Feeling of “heavy” legs	8.21 (±1.95)
Feeling of tired legs	7.79 (±2.17)
Bruising (hematomas)	7.63 (±2.31)
Feeling of tension in the legs	7.49 (±2.19)
Hypersensitivity to touch	7.32 (±2.42)
Swelling	7.04 (±2.41)
Pain in the affected areas	6.68 (±2.32)
Impairment in walking	6.45 (±2.78)
Pain in thighs:	6.34 (±2.5)
Pain in lower legs:	6.18 (±2.53)
Pain in arms:	5.33 (±2.72)
Feeling of cold in the legs	5.11 (±3.21)
Feeling of warmth in the legs	4.92 (±3.01)
Itching	4.78 (±3.00)
Pain in buttock:	4.65 (±2.62)
muscle cramps	4.63 (±2.86)
Skin complications	4.11 (±2.98)
Pain in belly:	4.00 (±2.72)

**Table 3 jcm-11-02836-t003:** Family history of lipedema or noticeably thick legs.

Family Member	Affected (*n* = 511)
mother	33.6%
grandmother (in general)	28.4%
grandmother maternal	15.4%
sister	14.6%
grandmother paternal	10.3%
aunt paternal	9.8%
aunt maternal	9.2%
daughter	2.3%

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
