# Peer review of "Characteristics and Patient Reported Outcome Measures in Lipedema Patients—Establishing a Baseline for Treatment Evaluation in a High-Volume Center"

_jcm, 2022, doi:10.3390/jcm11102836_

Round 1

Reviewer 1 Report

Interesting paper.

I have some minor comments:

p2-4: Can you change the lay-out of table 1 and 2? Not clear, disorderly, add legend with explanation of abbreviations and scoring

Table 1: Why mentioning 'added by Herbst'

line 59: intermittent

line 72-74: please re-phrase

line 118-119: patient exclusion criteria should be mentioned in the patient selection and not in the data collection

line 272: transition is weird, please add another sentence in between the two sections to make it clear that you are now talking about an intervention option

line 278: 'be' instead of bee

Reviewer 2 Report

It's a very original Paper that deals with many of the aspects of Illness, with competence and with total updating of data currently known in Literature. It is an important and innovative scientific and cultural contribution.

I suggest to deep the Family aspects/hereditary with the anamnesis collection.

-In the casuistry, the Authors could have better highlighted the number of normal weight patients (BMI between 20 and 25) compared to obese

-It does not emerge whether some of the patients had undergone bariatric surgery (and possibly, in this case, what type of surgery) or previous Liposuction surgery

-The Authors could have shown at least two of three of pedigrees because in our experience there is always at least one clinical case present in the family

-In reconstructing the pedigree of the individual cases, the Authors could have highlighted the forms transmitted by the patient's 'father' (father can transmit but never suffer from the disease) simply by adequately reconstructing the female family cases of the paternal family.
